# Next-Generation Computationally Designed Influenza Hemagglutinin Vaccines Protect against H5Nx Virus Infections

**DOI:** 10.3390/pathogens10111352

**Published:** 2021-10-20

**Authors:** Ivette A. Nuñez, Ying Huang, Ted M. Ross

**Affiliations:** 1Center for Vaccines and Immunology, University of Georgia, Athens, GA 30602, USA; ivette.a.nunez@gmail.com (I.A.N.); yhuang0@uga.edu (Y.H.); 2Department of Infectious Diseases, University of Georgia, Athens, GA 30602, USA

**Keywords:** COBRA, influenza, H5N1, avian influenza vaccine, H5Nx vaccine

## Abstract

H5N1 COBRA hemagglutinin (HA) sequences, termed human COBRA-2 HA, were constructed through layering of HA sequences from viruses isolated from humans collected between 2004–2007 using only clade 2 strains. These COBRA HA proteins, when expressed on the surface of virus-like particles (VLP), elicited protective immune responses in mice, ferrets, and non-human primates. However, these vaccines were not as effective at inducing neutralizing antibodies against newly circulating viruses. Therefore, COBRA HA-based vaccines were updated in order to elicit protective antibodies against the current circulating clades of H5Nx viruses. Next-generation COBRA HA vaccines were designed to encompass the newly emerging viruses circulating in wild avian populations. HA amino acid sequences from avian and human H5 influenza viruses isolated between 2011–2017 were downloaded from the GISAID (Global Initiative on Sharing All Influenza Data). Mice were vaccinated with H5 COBRA rHA that elicited antibodies with hemagglutinin inhibition (HAI) activity against H5Nx viruses from five clades. The H5 COBRA rHA vaccine, termed IAN8, elicited protective immune responses against mice challenged with A/Sichuan/26621/2014 and A/Vietnam/1203/2004. This vaccine elicited antibodies with HAI activity against viruses from clades 2.2, 2.3.2.1, 2.3.4.2, 2.2.1 and 2.2.2. Lungs from vaccinated mice had decreased viral titers and the levels of cellular infiltration in mice vaccinated with IAN-8 rHA were similar to mice vaccinated with wild-type HA comparator vaccines or mock vaccinated controls. Overall, these next-generation H5 COBRA HA vaccines elicited protective antibodies against both historical H5Nx influenza viruses, as well as currently circulating clades of H5N1, H5N6, and H5N8 influenza viruses.

## 1. Introduction

The H5 highly pathogenic avian influenza viruses from the genetic clade 2.3.4.4 emerged in China in 2010–2011. These viruses were detected in birds in more than 24 countries in the eastern hemisphere [1]. As of February 2020, the World Health Organization (WHO) reported 24 cases of human infections [2] by viruses of the H5N6 subtype, which is the only 2.3.4.4 clade H5 influenza virus that has infected people [3]. The newest cases of H5N6 virus infections in humans both occurred in China in September 2017 and recent infections in people with viruses that had HA proteins that were phylogenetically distinct from one another [4]. In 2014, H5N8 Eurasian subtypes was detected in U.S. Washington state in captive falcons, wild birds, and poultry [5,6]. These viruses spread across the central regions of North America devastating the poultry industry [6]. The clade 2.3.4.4 viruses from Africa and Europe were primarily of the H5N8 subtype, whereas those viruses isolated in Asia are in the H5N6 subtype and those strains isolated in the United States are classified as H5N2 isolates [4]. Clade 2.3.4.4 H5Nx viruses isolated from Africa and Europe are primarily of the H5N8 subtypes and those found in Asia are H5N6 subtype [1]. The H5N8 viral subtype caused an outbreak in 2014 in South Korea leading to a distinction of two different H5N8 virus subgroups [6,7,8]. Group A comprises a set of H5N8 isolates and is referred to as the intercontinental group A (icA) group. The icA H5 viruses further evolved into 3 different subgroups, icA1, icA2, icA3 [6].

Highly pathogenic H5N8 avian influenza viruses (AIV) were responsible for outbreaks in Taiwan and South Africa, as well as central and eastern Europe in 2020. In August 2020, H5N8 influenza viruses caused outbreaks in Russia in both poultry and wild fowl and the affected regions extending from Kazakhstan to several western European countries. H5N8 influenza virus outbreaks arose in poultry and/or wild birds in Israel, Japan, and South Korea resulting in millions of poultry being slaughtered. Phylogenetically, these H5N8 influenza viruses belong to clade 2.3.4.4b [9], which was previously a minor clade of avian influenza viruses [4,9]. In February 2021, the Russian Federation notified the WHO that it detected the first infections in people with H5N8 AIVs, all from poultry workers in Astrakhan Oblast, Russia [10].

Avian influenza H5Nx viruses from clade 2.3.4.4 caused a different pathogenicity in chicken, duck and mammal species [11,12]. Ducks infected with H5N6 and H5N8 virus have viral growth in the lung, spleen, kidneys and brain [12]. However, mortality in the duck species is variable, infection in ducks with H5N6 or H5N8 caused death in one animal per group per virus. Ducks infected with a H5N1 virus resulted in 100% mortality [12]. The pathogenicity induced by clade 2.3.4.4 influenza viruses in chickens was more severe and 100% of the birds die at 4–5 days post-infection following H5N6 and H5N8 influenza virus infection [12]. BALB/c mice that are infected with H5N6 influenza viruses with increased binding affinities to α-2,3-sialic acids die from infection and mice infected with H5N6 influenza viruses with binding affinities to α-2,6-sialic acids had no pathogenic effects [11]. Ferrets infected with H5N6 influenza viruses have a range of pathological effects compared to HPAI H5N1 influenza virus infections. Ferrets infected with H5N6 influenza viruses had no signs of morbidity, except for fever 1-day post-infection and they replicated efficiently in the lungs and spleens of challenged ferrets [13]. Ferrets did, however, easily transmit the virus through direct contact naïve cage mates, through airborne droplets, whereas H5N1 influenza infected ferrets did not transmit by either route [13,14].

Viruses from clade 2.3.4.4 have binding affinities to both α-2,3 sialic acids and α-2,6 sialic acid residues [11]. Mutations at amino acid position S123P, I151T, T156A and a deletion at position 125 in the HA domain are associated with receptor binding alterations. The HA of avian influenza viruses can be altered to bind to human α-2,6-linked sialic acids by introduction of a single amino acid mutation into the RBS [9]. Also, amino acid sites 182, 222, 223 and 224 are important sites for avian viruses to bind to α-2,6 sialic acids. HPAI viruses that are of H5Nx subtype in clade 2.3.4.4 bind to both sialic acid subtypes. These mutations increase the possibility of these influenza viruses becoming a human transmissible agent.

Viruses in the genetic clade 2.3.4.4 are genetically and antigenically distinct from other clade 2 viruses [15]. The H5N6 influenza viruses contain reassortments from multiple viruses, including internal genes that originate from H5N1, H6, H3, and H9N2 [16]. Viruses with internal genes from H9N2 have been responsible for 12 human infections during 2015–2016 [16]. The H5N6 influenza viruses also have internal genes from HPAI H5N1 viruses from clade 2.3.2.1c [7]. HA genes from clade 2.3.4.4 H5Nx viruses can be divided into four subclades designated I-IV [7]. The H5N6 viruses are found in subclades I and II, H5N2 and H5N8 viruses are found in subclade III and subclade IV are mainly H5N8 [7]. These major shifts in the HA, NA and internal gene segments have resulted in H5N6 viruses that do not elicit antibodies that are cross reactive against strains in the H5N1 subtypes [17]. Chickens that are vaccinated with recombinant vaccines are protected from the pathogenicity against these H5N6 viruses but are still able to transmit the virus to other poultry [13]. Poultry vaccinations, therefore, need to be re-evaluated in order to provide protection against these dominate H5N6 circulating strain.

The prevalence of H5N6 and H5N8 circulating in wild waterfowl populations has increasingly become an issue for the health of poultry farmers. Viruses from clade 2.3.4.4 can reassort with NA gene segments that are naturally found in avian species and have an increasing tendency towards binding to sialic acid receptors more commonly found in the upper respiratory tract of humans. These features of the H5Nx viruses have further increased the potential of these viruses to cross over into the human population. Along with reassortant events, the HA mutational rate is also problematic, as observed by the phylogenetic branching and lack of HAI titers against reference strains. In the last two years, two major viral clades have been circulating in the wild waterfowl populations, clade 2.3.4.4 and 2.3.2.1. [2]

Mandatory vaccination of poultry was established in Guangdong province in China using an inactivated influenza virus vaccine [18]. This vaccine regimen decreased the prevalence of H7N9 influenza virus circulation in live poultry markets, however, circulation of H5N6 viruses continued and increased in antigenic diversity compared to the vaccine strain [18]. In 2018, the Chinese Government and the WHO approved a new A/Guangdong/18F020/2018 candidate vaccine virus [18]. However, vaccine escape mutants are still a risk for the animal and human populations. Reference sera generated by the WHO revealed that reference antigens A/Sichuan/26221/2014, A/Hubei/29578/2016 and A/Fujian-Sanyuan/21099/2017 do not generate antibodies against the A/Guangdong/18SF020/2018 vaccine strain [19].

Although clade 2.3.4.4 H5Nx influenza viruses are the dominant circulating strains in wild waterfowl and poultry populations, outbreaks of H5N1 viruses from clade 2.3.2.1 have also been reported. In 2015, viruses from clade 2.3.2.1 were the cause of the majority of the H5N1 outbreaks since 2011. Similar to the 2.3.4.4 viruses, clade 2.3.2.1 H5 viruses have further diversified into five separate subclades [20]. Since 2010, the HA of clade 2.3.2.1 viruses have spread over provinces in Vietnam in early 2014. Specifically, since the first outbreak in 2010 in chickens and ducks, viruses from the subclade 2.3.2.1c are predominant in southern Vietnam [21]. These compounding factors further exacerbate the need for a pandemic vaccine for both domestic poultry and the at-risk human population that spans multiple viral H5Nx clades.

## 2. Materials and Methods

### 2.1. Next-Generation Computationally Optimized Broadly Reactive Antigens (COBRA) Design

Next generation computationally optimized broadly reactive antigens (COBRA) H5 HA antigen were generated through a consensus sequence alignment of H5NX HA sequences from human and avian isolates. Sequences were downloaded through the GISAID database based on area, date of submission and the species of isolation. These sequences were then organized and used to generate multiple consensus sequences in order to capture the repeated and unique H5 epitopes. The COBRA approach used 10–20 primary consensus sequences isolated over a 4–5 year time frame using the sequences taken from 2011–2015, 2012–2016, 2013–2017) and one 5-year-long span (2011–2016). The HA sequence was downloaded into Geneious (San Diego, CA, USA) and aligned using Muscle alignment. The HA1 fragment of each HA sequence was extracted to produce the unique HA sequences. The AAs 17–340 were extracted and were then imported into a new file for re-alignment. The remaining 322 AA were used to create the COBRA HA1 sequence (Appendix A). These sequences were used to generate a phylogenetic tree (Figure 1) and were then condensed based upon identity and on the tree. Sequences that were condensed had no more that 2.5% difference and no ambiguities (X amino acid). Each primary sequence was labeled to represent the original sequences that were used in each primary consensus sequence. These primary consensus sequences were further combined into another phylogenetic tree and were combined to create unique sequences with no ambiguities. Over 50 sequences were generated using this method, but only eight were chosen due to their unique AA sequence and their placement on the phylogenetic tree. Sequences that were clustered too closely together with wild-type sequences and were not found to be closely associated with the root were ruled out. Each segment was blasted to confirm its uniqueness. The leader sequences (first 17 AA) were taken from a wild-type virus that was closely related to the unique COBRA virus. This was done to ensure the sequence would be properly localized in the cell. The final 8 sequences were generated by Genewiz (South Plainfield, NJ, USA) into out acceptor vector plasmid Zeo+ pcDNA3.1 (Thermo Fisher Scientific, Waltham, MA, USA).

### 2.2. Recombinant Protein Production

Each wild-type and COBRA recombinant HA protein was purified as described as in Ecker et al. (2020). Briefly, the HA gene cassettes expressing wild-type or COBRA HA recombinant protein from the H5NX subtype were cloned into mammalian DNA expression plasmid pcDNA 3.1/Zeo(+)vector (Thermo Fisher Scientific) and were synthesized by Genewiz (South Plainfield, NJ, USA). The plasmid was transformed into the Top 10 bacterial cell line and was purified using Zympure maxi-prep. The HA1 fragment, which contained a KPNI site was removed from the plasmid and was moved into an acceptor vector containing the Hu-CO2 HA2 domain. The final gene of the HA protein contained an extracellular domain that was terminally fused with the trimeric domain of T4 fibritin, an AviTag sequence and a hexahistidine affinity tag for purification [22]. Each DNA plasmid containing either wild-type or COBRA antigens were transiently transfected into Expi293F HEK suspension cell line (Thermo Fisher Scientific) and was allowed to incubate for 72 h at 37 °C (5% CO2). Supernatants were collected and were tested for protein expression through BCA and Western blot (His tag antibody). The cells were then pelleted down and the supernatant was purified for protein collection. Soluble HA protein was purified via AKTA Pure System using HisTrap columns following the manufacturers protocol. Eluted fractions were pooled and purified, protein concentration was tested though anti-HIS tag antibody (Biolegend, San Diego, CA, USA) using SDS-PAGE and Western blot [23].

### 2.3. Viruses

Viruses were obtained through the Influenza Reagents Resource (IRR) and passaged once in embryonated chicken eggs as per the instructions provided by the WHO [24]. Virus lots were tittered with horse erythrocytes and made into aliquots for single-use applications. The H5NX vaccine panel includes the following reassortant PR8 (2:6) viral strains containing internal genes from the mouse adapted strain A/Puerto Rico/8/1934: A/Vietnam/1203/2004 (Vn/04), A/Whooper swan/Mongolia/244/2005 (ws/Mo/05), A/Anhui/1/2005 (An/05), A/Egypt/321/2007 (Eg/07), A/chicken/Vietnam/NCVD-16/2008 (ck/Vn/08), A/Hubei/1/2010 (Hu/10), A/Egypt/N03072/2010 (Eg/10), A/Guizhou/1/2013 (Gu/13), A/Sichuan/26221/2014 (Si/14), A/gyrfalcon/Washington/41088-6/2014 (gyr/WA/14).

### 2.4. Mouse Studies

BALB/c mice (female, 6- to 8-weeks-old) were purchased from The Jackson Laboratory (Bar Harbor, ME, USA) and were housed in microisolator units and fed ad libitum. Mice were handled in accordance with UGA IACUC protocols and were cared for under the U.S. Department of Agriculture guidelines for laboratory animals. Mice were humanely euthanized in case of weight loss ≥25% of the original weight. After the mice were acclimated for 7 days, they were bled to ensure all were immune naïve prior to vaccination. After naïve mice were confirmed, mice were vaccinated using 5 µg of recombinant protein formulated with an oil-in-water nano-emulsion adjuvant AddaVax™ according to the manufacturer’s protocols. Mice were vaccinated three times at a 4-week interval to obtain appropriate antibody response (*n* = 10). Four weeks following the last vaccination, mice were intranasally infected with 2 × 10^7^ pfu of recombinant A/Sichuan/26621/2014 virus and 1 × 10^7^ pfu of A/Vietnam/1203/2004-PR8 reassortant virus. Mice were briefly anesthetized in an isoflurane chamber and were intranasally inoculated with 50µL of virus. The mice were allowed to recover and were monitored 2× daily for weight loss, clinical signs and mortality for up to 14 days.

### 2.5. Hematoxylin and Eosin (H&E) Staining

To assess the viral replication and pathological effect of infection, mice (n = 3) were euthanized 3 days post infection. The right lung lobes were taken for viral plaques and the incision was clamped with a hemostat, a 22 gauge needle was then used to puncture the apex of the heart and sterile PBS was perfused throughout the mouse for 2–3 min. After the blood was efficiently removed from the lungs, 10% formalin was then perfused to fix the left lobes. Lungs were removed and placed into formalin for 1 week prior to paraffin embedding. Mouse lungs were embedded in paraffin and were cut using a Lecia microtome. Transverse 5 µm sections were placed onto Apex superior adhesive glass slides (Leica biosystem Inc., Lincolnshire, IL, USA) which were coated for a positive charge. and were processed for H&E staining. Sections were deparaffinized in xylene and hydrated using different concentrations of ethanol (100%, 95%, 80% and 75%) for 2 min each. Deparaffinized and hydrated lung sections are stained with hematoxylin (Millipore sigma, Burlington, MA, USA) for 8 min at RT, differentiated in 1% acid alcohol for 10 s, and then counterstained with eosin (Millipore sigma, Burlington, MA, USA) for 30 s, slides were dehydrated with 95% and 100% ethanol, cleared by Xylene, and mounted using Permount^®^ mounting media (Thermo Fisher scientific, Waltham, MA, USA).

### 2.6. Immunohistochemistry Staining

The deparaffinized and hydrated lung tissue sections were subjected to antigen retrieval by sub-boiling in 10 nm sodium citrate buffer at pH = 6 for 10 min and then incubated in 3% fresh made hydrogen peroxide for 10 min to inactivate endogenous peroxidase at room temperature. The lung sections were blocked with 5% horse serum in PBS, incubated with mouse Influenza A Nucleoprotein monoclonal antibody at 1:1000 dilution (Bio-Rad, Hercules, CA, USA) overnight at 4 °C, and then incubated with biotinylated goat-antibody mouse IgG H&L (Abcam, Burlington, MA, USA) at 1:2000 dilution for 1 h at RT. The avidin-biotin-peroxidase complex (VectStain Standard ABC kit) (Vector Laboratories, Burlingame, CA, USA) was used to localize the biotinylated antibody, and DAB (Vector Laboratories, Burlingame, CA, USA) was utilized for color development. Sections were then counterstained with hematoxylin, and then mounted using Permount^®^ mounting media (Thermo Fisher scientific, Waltham, MA, USA). Images were obtained by Aperio digital slide scanner AT2 (Leica biosystem, Lincolnshire, IL, USA).

### 2.7. Plaque Assays

Viral titers were determined in BALB/c mice using a plaque-forming assay as previously described [25,26,27,28,29] using 1 × 10^6^ Madin-Darby Canine Kidney (MDCK) cells. Mice were euthanized (*n* = 3/group) 3 days post-infection, lungs were taken and snapped frozen and kept at −80 °C until processing. Lungs were diluted (10^0^ to 10^6^) and overlaid onto confluent MDCK cell layers for 1 h in 200 µL of DMEM supplemented with penicillin-streptomycin. Cells were washed after 1-h incubation and DMEM was replaced with 4 mL of L15 and 2.4% Avicel (FMC BioPolymer; Philadelphia, PA, USA) (1:1). Cells were incubated for 72 h at 37 °C with 5% CO_2_. Avicel and L15 media was removed and the samples were washed twice with sterile PBS, then cells were fixed with 10% buffered formalin and stained for 15 min with 1% crystal violet. Cells were washed with tap water and allowed to dry. Plaques were counted and the plaque forming units calculated (PFU/mL)

### 2.8. Hemagglutination-Inhibition (HAI) Assay

The hemagglutinin-inhibition assay (HAI) assay was used to assess receptor-blocking antibodies to the HA protein to inhibit agglutination of horse erythrocytes. The protocol was taken from the CDC laboratory influenza surveillance manual. To inactivate non-specific inhibitors, mouse sera was treated with receptor destroying enzyme (RDE, Denka Seiken, Co., Tokyo, Japan) prior to being tested. Three parts of RDE were added to one-part sera and incubated overnight at 37 °C. The RDE was inactivated in 56 °C for 30 min and, when cooled, six parts of sterile PBS was added to the sera and was kept at 4 °C until use. RDE-treated serum was two-fold serially diluted in v-bottom microtiter plates. 25 µL of virus at 8 HAU/50 µL was added to each well (4 HAU per 25 µL). Plates were covered and incubated with virus for 20 min at room temperature before adding 1% horse red blood cells (HRBC) (Lampire Biologicals, Pipersville, PA, USA) in PBS. Red blood cells were washed and stored at 4 °C and used within a week of preparation. The plates were mixed by agitation and covered, and the RBCs were allowed to settle for 1 h at room temperature. HAI titer was determined by the reciprocal dilution of the last well which contained non-agglutinated RBC. Negative and positive serum controls were included for each plate. All mice were negative (HAI < 1:10) for pre-existing antibodies to currently circulating human influenza viruses prior to vaccination.

### 2.9. P-Epitope/P-Sequence Analysis

In order to assess the antigenic distances between the HA sequences used in the vaccines and the HA sequences used in the target strains, a $P_{-sequence}$ analysis was performed on the vaccine and virus strain and used to calculated antigenic distances. The epitopic value was calculated by the number of amino acid changes divided by the number of amino acids located in a specific antigenic epitope. A linear regression analysis was performed in Prism in order to determine a correlation between HAI titers and p-epitope.
(1)$$P_{-sequence}\;=\;\frac{\begin{array}{c} Number\;of\;substitiutions\;in\;the\;\\ HA1\;RBS\;domain\;of\;hemagglutinin \end{array}}{\begin{array}{c} Total\;number\;of\;amino\;acids\;in\;the\;\\ HA1\;RBS\;doamin\;of\;hemagglutinin \end{array}}$$

## 3. Results

### 3.1. Design and Characterization of COBRA H5 Hemagglutinin (HA) Vaccines

COBRA H5 HA vaccines were designed using H5Nx HA sequences downloaded from GISIAD (Appendix A). A multilayered consensus building approach was applied to 4524 A(H5Nx) HA amino acid sequences collected from 2011 to 2017, that resulted in the generation of eight unique next-generation HA sequences (Table 1). All next-generation H5 HA amino acid sequences were unique and did not match the amino acid sequence of any HA in a wild-type A(H5Nx) isolate. Vaccines were phylogenetically spread across multiple viral clades (Table 1).

Each wild-type and COBRA H5 HA protein was expressed in mammalian cell lines purified over a nickel column using the carboxyl-terminal 6× HIS-tag. These purified HA proteins were used as immunogens to vaccinate BALB/c mice (*n* = 8; 6–8 weeks of age) at day 1 and boosted at day 28. Along the 8 vaccines used, groups of mice were vaccinated with one of four wild-type rHA (WS/05, Sich/14, Gry/WA/14, ck/Egypt/17) (Figure 2). Mice were vaccinated with a traditional H5 COBRA HA antigen, Hu-CO 2, as a positive control and a mock vaccinated mice were used as a negative control. Sera collected at day 42 post-vaccination was tested for HAI activity against a panel of five viruses, WS/05 (2.2), Gu/13 (2.3.4.2), Hu/10 (2.3.2.1), Si/14 (2.3.4.4) and gy/WA/14 (2.3.4.4). These viruses represented four distinct H5N1, H5N6, and H5N8 Clade 2 viruses isolated between 2005–2014 (Figure 1).

### 3.2. Vaccines Elicit Antibodies with Hemagglutination-Inhibition Activity

Mice vaccinated with wild-type HA antigens elicited antibodies with HAI activity against some, but not all, H5Nx viruses in the panel (Figure 2). The two HA vaccines based upon 2.3.4.4 viruses elicited antibodies with high HAI activity against the 2.3.4.4. viruses and Gz/13 (Figure 2A,B), but not the clade 2.2 based H5 viruses. In contrast, WS/05 HA vaccinated mice had antibodies with HAI activity against Hu/10 and WS/05 viruses (Figure 2D) that were similar to the antibodies elicited by the traditional H-CO2 HA vaccine (Figure 2E) [30].

Mice vaccinated with the next-generation H5 COBRA HA antigens elicited antibodies with HAI activity against different sets of H5Nx viruses (Figure 3). IAN-3 and IAN-6 HA vaccinated mice elicited antibodies that had similar HAI activity as antibodies elicited by the wild-type clade 2.3.4.4 HA proteins from Si/14 and gy/WA/14 (Figure 3C,F). Mice vaccinated with IAN-2 HA did not elicit antibodies with HAI activity against any of the viruses in the panel (Figure 3B). Overall, only three out of the eight next generation H5 HA vaccines, IAN-4, IAN-7 and IAN-8 HA vaccines elicited antibodies with HAI activity against H5 viruses in all four clades. All mock vaccinated mice were serologically negative to the viruses in the panel.

To further explore IAN-4, IAN-7 and IAN-8 HA vaccines in more depth, these vaccines were used to vaccinate a new set of mice to determine the protective efficacy against H5Nx challenges. BALB/c mice (*n* = 16) were vaccinated and then boosted twice at days 28 and 56 and sera collected at day 70. Each vaccine elicited a distinct antibody profile against one of the four additional viruses in the vaccine panel (Figure 4). Mice vaccinated with the IAN-4 HA vaccine did not develop robust HAI activity against the two viral challenge strains, VN/04 or Si/14 (Figure 4B). Mice vaccinated with IAN-7 HA had high HAI antibody titers against clade 2.3.4.4 viruses (Figure 4A, Appendix A) and moderate HAI activity against An/05 and Eg/07 (Figure 4). Mice vaccinated with IAN-8 HA developed HAI antibodies to an average titer of 1:80 across all five viruses tested in the panel (Figure 4 and Appendix A). These levels of HAI antibodies were similar to mice that were vaccinated with the VN/04 HA, as well as IAN4, IAN-7 and IAN-8 HA. In comparison, the original H-CO2 HA vaccines did not induce antibodies with HAI activity against the ck/VN/08 virus.

### 3.3. Viral Challenge

Mice were challenged with either VN/04 or Si/14 at day 56 post-vaccination (Figure 5). Mice lost ~75% of their body weight by day 7 post-infection, regardless of the virus used for infection. All vaccinated mice infected with SI/14 survived challenge (Figure 5C), and mice vaccinated with the SI/14 or gry/WA/14 HA maintained the same average weight for the 10 days of observation. Mice vaccinated with IAN-7 HA actually gained weight following the challenge (Figure 4A). Mice vaccinated with the other COBRA HA or wild-type HA antigens lost between 7–10% body weight by day 3 and then recovered to full body weight by day 10 post-infection. In contrast, only one mouse vaccinated with IAN-7 HA survived VN/04 challenge (Figure 4D). Seventy percent of mice vaccinated with SI/14, gry/WA/14 or IAN-4 HA survived challenge with VN/04, but they lost an average 10–12% of their original body weight by day 5 post-infection. This weight loss was statistically the same as mice vaccinated with VN/04, IAN8, or Hu-CO-2 HA vaccinated mice (Figure 4B), which all survived VN/04 challenge (Figure 4D).

Mock vaccinated mice had ~5 × 10^3^ pfu lung viral titers regardless which of the two H5Nx viruses was used for challenge (Figure 6). All vaccinated mice had no SI/14 virus detectable in lung tissue at day 3 post-infection (Figure 6B). In contrast, mice vaccinated with VN/04 or IAN-8 HA vaccines had an average of 10 pfu viral lungs titers collected 3 days post-infection with VN/04 virus (Figure 6A). Mice vaccinated with the other 4 HA vaccines had low to moderate lung viral titers (50–200 pfu) from lungs collected day 3 post-infection with VN/04 (Figure 6B).

### 3.4. Histopathology

Lungs collected on day 3 post-infection were analyzed for histopathology by H&E staining (Figure 7 and Figure 8) and immunohistochemistry (IHC) (Figure 9 and Figure 10) for detection of influenza virus NP nucleoprotein. Mice vaccinated with IAN-4, IAN-7, VN/04 HA and the mock control had the highest amounts of cellular infiltrates and inflammation (Figure 7A,B,F,G) following Si/14 virus challenge. However, mice vaccinated with IAN-8 HA or the homologous Si/14 HA control had fewer stained infiltrating cells that were similar to the unchallenged mock control lungs (Figure 7C,E,H). The amount of inflammation in the lungs of IAN-4 HA vaccinated mice correlated with the increased amounts of virus detected in these lungs (Figure 7A). Mice challenged with VN/04 virus also had high levels of lung inflammation in IAN-4 HA vaccinated and in Si/14 HA vaccinated mice (Figure 8A,E). Inflammation was inhibited in the lungs of mice that were vaccinated with either IAN-7, IAN-8, Hu-CO2 or VN/04 HA compared to the mock vaccinated control challenged lungs (Figure 8B–D,F–G). VN/04 influenza virus infection induced less lung inflammation than Si/14 challenged mice.

The Si/14 virus was easily detected in the epithelial cells in IAN-4, IAN-7, VN/04 HA vaccinated mice by IHC (Figure 9A,B,F), which was comparable to the level of virus detected in the control unvaccinated mice (Figure 9G). Mice vaccinated with IAN-8, Hu CO2 or Si/14 HA vaccines had the fewest cells staining positive for viral NP (Figure 9C–E), which was consistent with the amount of inflammation observed by H&E staining (Figure 7C–E). Mice challenged with VN/04 virus had high levels of viral staining on the epithelial cells in mice vaccinated with IAN-4, IAN-7, Si/14 HA or mock control unvaccinated groups (Figure 10A,B,E,G). IAN-8 HA vaccinated mice had lungs with IHC staining levels that were similar to mock unchallenged lungs with low levels of viral NP staining (Figure 9C,H). Lastly, the lungs of mice vaccinated with VN/04 HA vaccines had viral NP stained epithelial cells, but increased cellular infiltrates were consistent with a lack of NP binding.

## 4. Discussion

As part of universal vaccine development, next-generation influenza virus vaccines not only should protect against seasonal IAV and IBV antigenic drift, but also against the emergence of novel strains and subtypes not currently circulating in the human population. Therefore, one goal is to generate an influenza virus vaccine that efficiently protects against emerging pandemic virus subtypes with multiple antigenic variants, such as H5Nx strains. In this study, COBRA HA designed vaccines were evaluated in a mouse model by comparing the elicited immune responses and protective efficacy to wild-type H5 HA vaccines. Three next-generation vaccines, IAN-4, IAN-7 and IAN-8 HA, were as efficient as the original H5 COBRA HA, H-CO-2 [26], at eliciting broadly reactive immune responses. All vaccinated mice were protected against a lethal challenge of Si/14 virus with no detectable viral lung titers at day 3 post-infection. When challenged with a lethal dose of VN/04 virus, mice vaccinated with IAN-8 rHA had little weight loss and decreased viral lung titers compared to IAN-4, IAN-7, and H-CO2 HA vaccinated mice.

The methodology to generate these next-generation H5 HA vaccines differs from the approach used to generate previous COBRA H5 HA antigens [31] Both traditional and next-generation H5 COBRA HA vaccines induced cross reactive antibodies against multiple H5 viral clades. However, the elicited HAI activity varied between vaccines, as well as the ability to prevent morbidity and mortality in vaccinated mice. The IAN-8 HA vaccine induced antibody titers against 9 out of the 10 H5 viruses derived from different clades. However, antisera collected from gy/WA/14 HA vaccinated had no HAI activity against the VN/04 virus, but ~71% of the vaccinated mice survived a lethal VN/04 viral challenge. The immune correlates of protection for H5 have not been as well established as the correlates for seasonal influenza viruses, however, in this study, the antibody titer elicited by the COBRA HA vaccines against a specific strain directly correlated with survival. This inconsistent correlation between low HAI titers and survival appears to be systemic for H5-specific vaccines. Previous studies have also discovered increased survival rates and decreased pathogenicity without detectable serum antibody titers [32,33]. This non-HAI protection has been attributed to stem-based antibodies [34] and/or anti-NA antibodies [35]. However, for this study, only rHA vaccines without NA were included. The role of cellular immunity cannot be ruled out since the use of oil-in-water adjuvants, such as those used in this study, enhances T-cell specific influenza virus vaccine immunity [36,37]. The role of cellular immunity induced by COBRA HA vaccination could be further analyzed.

Previously, mice vaccinated with VLP vaccines expressing the Hu-CO2 HA did not survive challenge and the collected antisera did not have HAI activity against the Si/14 virus. However, neutralizing antibody titers were detectable at higher concentrations [30] and the Hu-CO2 HA vaccinated mice survived a challenge. This may be due to increased antibody titers against HA specific epitopes, since these vaccines lack of NA and Gag_p24_ core proteins that are included in VLP vaccine formulations. The Hu-CO2 COBRA HA vaccine decreased the pathogenicity induced by the H5N6 influenza virus infection. IAN-8 HA vaccine appears to elicit immune responses against 2.3.2.1 H5Nx influenza viruses and future studies could be performed to demonstrate if IAN-8 HA induced immune responses protect against highly pathogenic variants of H5N6 or H5N8.

A p-epitope analysis of the three-vaccine strains was performed in order to examine the specific epitopes that were essential for HAI titer elicitation (Appendix A). P-epitope values were plotted against HAI titers for IAN-4, IAN-7 and IAN-7 HA against the HA in the two challenge VN/04 and Si/14 viruses. The specific H5 HA antigenic sites are not as well defined as the regions on seasonal influenza HA proteins, such as H1N1 and H3N2 subtypes. For this study, we defined the antigenic sites according to the review article by Velkov et al. [38] that described a broad guideline for antigenic sites that are designated as antigenic sites using an overlapping monoclonal antibody approach [38]. The amino acid 282 was also included in the analysis [39,40]. A modification in the methodology was performed in order to calculate the p-epitope value, as described previously [41,42,43,44]. The p-epitope was calculated using only the sites associated with a receptor binding site (RBS). This is an important tool that can be used when designing vaccines against pandemic strains of viruses.

When designing the next-generation COBRA HA vaccines, the goal was to produce unique HA proteins that encompassed not only the 2.3.4.4 clade, but also other clades that are circulating in avian species, specifically viruses in clade 2.3.2.1 that are circulating in Bangladesh, China and India [2]. These results strongly suggest that the IAN-8 rHA vaccine in combination with an oil-in-water adjuvant is a potential candidate for pre-clinical trials against clade 2.3.4.4 and 2.3.2.1 viruses, which are the dominant circulating clades in wild waterfowl populations throughout the world. Future studies using a ferret model of disease can be performed to ensure the protective efficacy of IAN-8 HA vaccination against a highly pathogenic strain of clade 2.3.4.4 virus and 2.3.2.1 virus.

## Figures and Tables

**Figure 1 pathogens-10-01352-f001:**
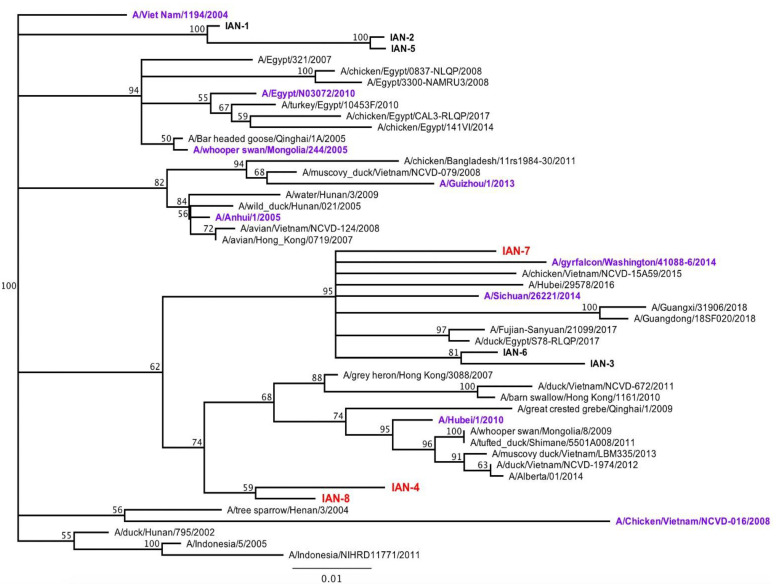
Phylogenetic Tree. Phylogenetic tree representing vaccine hemagglutinin (HA) sequence in comparison to wild-type viruses. Recombinant PR8 viruses are depicted in the tree as purple, where vaccines are listed in red. Tree was generated in Geneious software using Jukes Cantor Neighbor Joining Consensus, using VN/04 as an outgroup. Tree was resampled using Bootstrap method with 100 replicates.

**Figure 2 pathogens-10-01352-f002:**
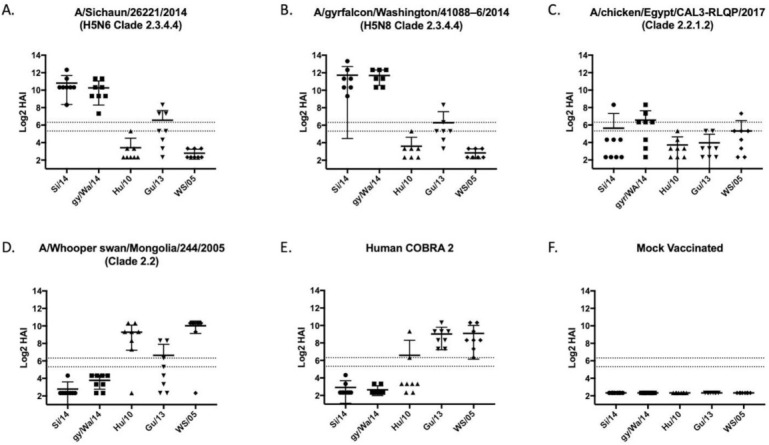
Wild-type and Hu-Co2 vaccine induced antibodies against five PR8-backbone viruses. Female BALB/c mice were vaccinated on a prime-boost-boost regimen using recombinant HA protein (rHA) using wild-type avian influenza sequences encoding (**A**) A/Sichuan/26221/2014, (**B**) A/gyrfalcon/Washington/41088-6/2014, (**C**) A/chicken/Egypt/CAL3-RLQP/2017; (**D**) A/whooper swan/Mongolia/244/2005; (**E**) Human COBRA 2; or were mock vaccinated (**F**). Serum from week 10 mice was taken to assess the immunological response against a hemagglutination-inhibition (HAI) panel of H5Nx viruses using horse erythrocytes. Antibody responses were calculated according to serum dilution, a value of 5 was given for a negative response. The two dashed lines represent HAI titers of 20 and 40. Each COBRA generated vaccine displayed a unique pattern of antibody response.

**Figure 3 pathogens-10-01352-f003:**
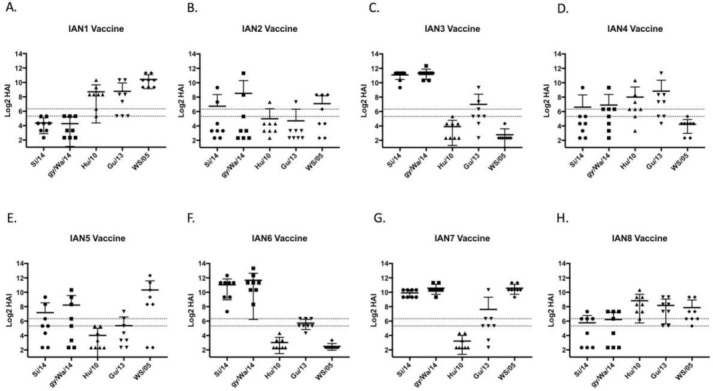
Next-gen vaccine induced antibodies against five PR8-backbone viruses. Female BALB/c mice were intramuscularly vaccinated with rHA proteins encoding next-generation COBRA sequences named (**A**) IAN-1; (**B**) IAN-2; (**C**) IAN-3; (**D**) IAN-4; (**E**) IAN-5; (**F**) IAN-6; (**G**) IAN-7; or (**H**) IAN-8. Serum from week 8 mice was taken to assess the immunological response against a HAI panel of H5Nx viruses using horse erythrocytes. Antibody responses were calculated according to serum dilution, a value of 5 was given for a negative response. The two dashed lines represent HAI titers of 20 and 40.

**Figure 4 pathogens-10-01352-f004:**
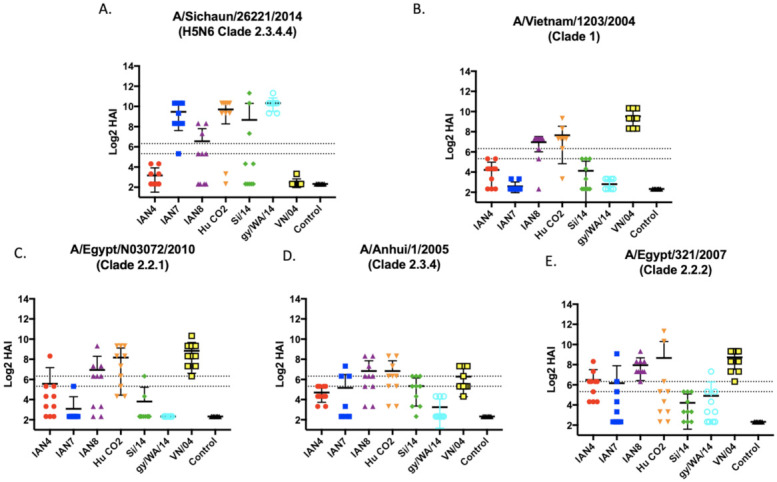
Hemmaglutinin inhibition antibodies elicited against the two viral challenge strains and 3 additional H5 viruses. Serum from mice taken 2 weeks following final vaccination in a prime-boost-boost model was tested against (**A**) Si/14 (**B**), VN/04 (**C**) A/Eg/10 (**D**) An/05, and (**E**) Eg/07. Antibody responses were calculated according to serum dilution, a value of 5 was given for a negative response. The two dashed lines represent HAI titers of 20 and 40. Viruses used were 6 + 2 recombinant, PR8-backbone isolates.

**Figure 5 pathogens-10-01352-f005:**
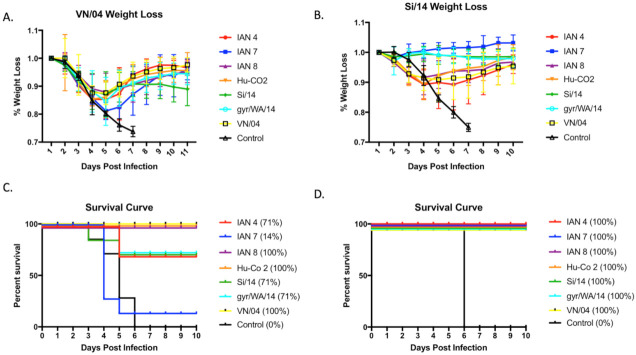
Weight loss data and survival curves from Vn/04 and Si/14 intranasal challenge. Four-weeks following the last boost vaccination, mice were intranasally challenged with (**A**) VN/04 or (**B**) Si/14 PR8 virus and weight loss was monitored daily for 10–11 days post-infection. Survival rates of mice vaccinated with either next-generation IAN vaccines or wild-type control were calculated for (**C**) VN/04 challenge and (**D**) Si/14 challenge.

**Figure 6 pathogens-10-01352-f006:**
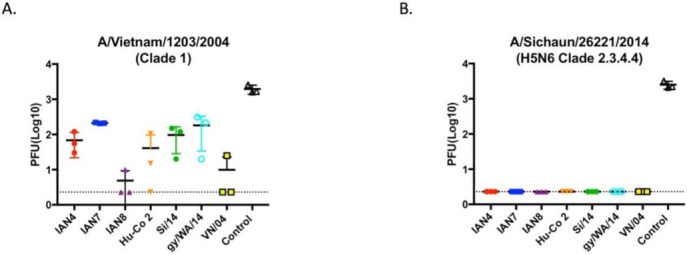
Viral lung titers obtained from mice 3-days post challenge. A subset of mice was randomly chosen to be sacrificed 3-days post intranasal infection for lung removal. Lung viral titers were calculated using influenza plaque assay on (**A**) Mice challenged with VN/04 reassortant PR8 virus and (**B**) Mice challenged with Si/14 reassortant PR8 virus. As expected, control mice that were not vaccinated had the highest titers compared to vaccinated mice. All vaccinated mice that were challenged with Si/14 had no viral lung titers. This was consistent with the lack of mortality in these groups.

**Figure 7 pathogens-10-01352-f007:**
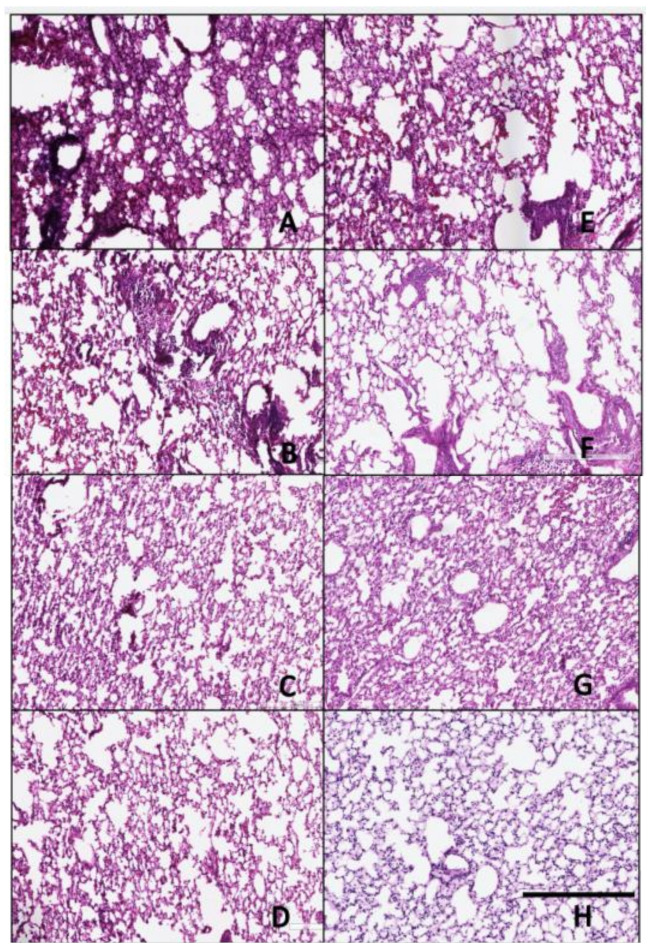
Histopathology of lungs taken 3 days following viral infection of Si/14 virus. Lungs were taken from euthanized mice and fixed in formalin for 1 week prior to examination. Hematoxylin and eosin staining (H&E) was performed on 5 µm lung slices to determine pathology and cellular infiltrates in vaccinated and non-vaccinated groups. (**A**) IAN-4, (**B**) IAN-7, (**C**) IAN-8, (**D**) Hu-CO2, (**E**) Si/14, (**F**) VN/04, (**G**) control infected (**H**) control mock (not infected). Bar represents 300 µm.

**Figure 8 pathogens-10-01352-f008:**
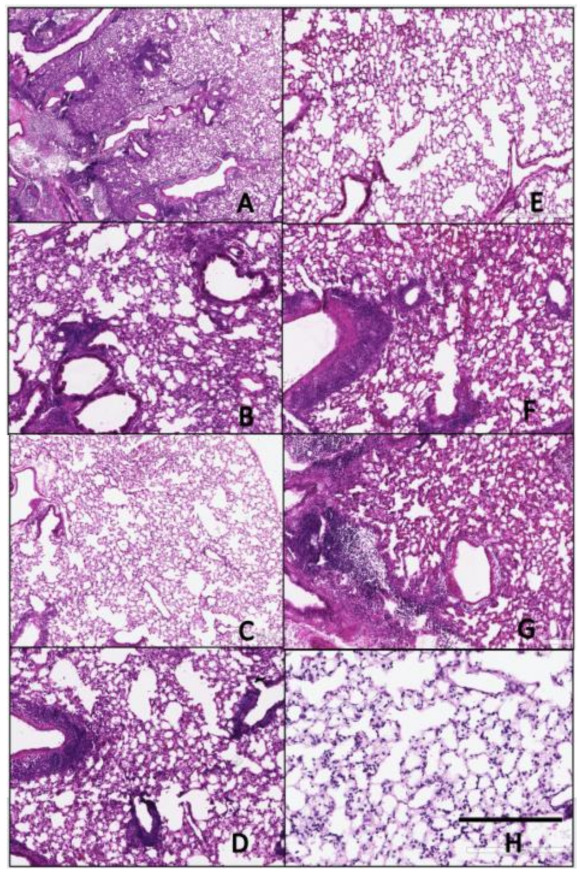
Histopathology of lungs taken 3 days following viral infection of VN/04 virus. Lungs were taken from euthanized mice and fixed in formalin for 1 week prior to examination. Hematoxylin and eosin staining (H&E) was performed on 5 µm lung slices to determine pathology and cellular infiltrates in vaccinated and non-vaccinated groups. (**A**) IAN-4, (**B**) IAN-7, (**C**) IAN-8, (**D**) Hu-CO2, (**E**) Si/14, (**F**) VN/04, (**G**) control infected (**H**) control mock (not infected). Bar represents 300 µm.

**Figure 9 pathogens-10-01352-f009:**
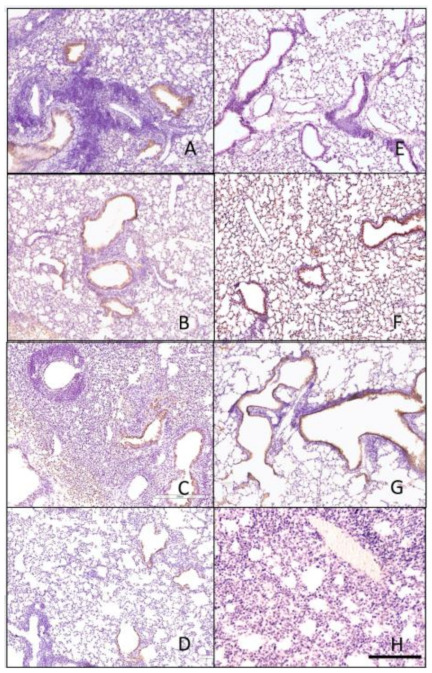
Immunohistochemistry of lungs taken 3 days following viral infection of Si/14 virus. Lungs were taken from euthanized mice and fixed in formalin for 1 week prior to examination. Immunohistochemistry (IHC) was performed on 5 µm lung slices to determine pathology and cellular infiltrates in vaccinated and non-vaccinated groups. (**A**) IAN-4, (**B**) IAN-7, (**C**) IAN-8, (**D**) Hu-CO2, (**E**) Si/14, (**F**) VN/04, (**G**) control infected (**H**) control mock (not infected). Bar represents 300 µm.

**Figure 10 pathogens-10-01352-f010:**
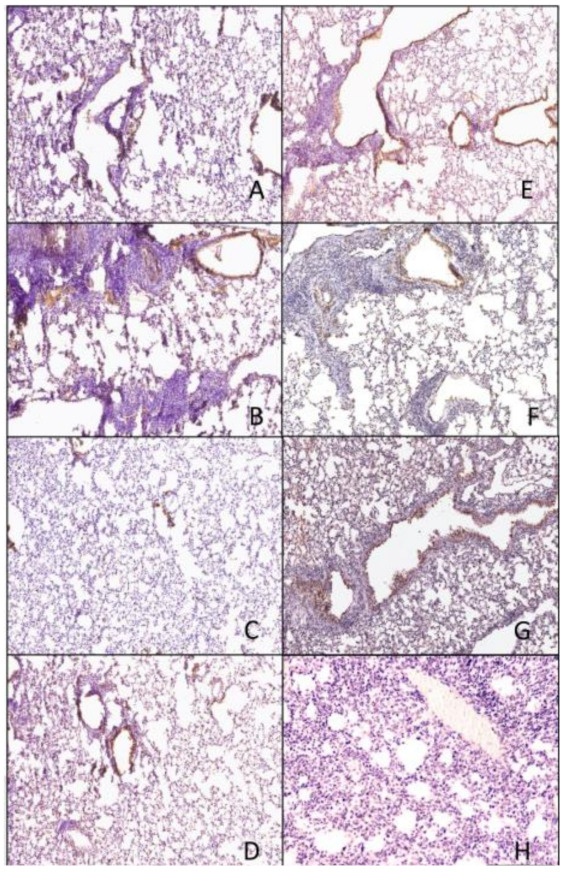
Immunohistochemistry of lungs taken 3 days following viral infection of VN/04 virus. Lungs were taken from euthanized mice and fixed in formalin for 1 week prior to examination. Immunohistochemistry (IHC) was performed on 5 µm lung slices to determine pathology and cellular infiltrates in vaccinated and non-vaccinated groups. (**A**) IAN-4, (**B**) IAN-7, (**C**) IAN-8, (**D**) Hu-CO2, (**E**) Si/14, (**F**) VN/04, (**G**) control infected (**H**) control mock (not infected). Bar represents 300 µm.

**Table 1 pathogens-10-01352-t001:** Vaccines were phylogenetically spread across multiple viral clades.

Group	Vaccine	Mice
1	IAN-1	8
2	IAN-2	8
3	IAN-3	8
4	IAN-4	8
5	IAN-5	8
6	IAN-6	8
7	IAN-7	8
8	IAN-8	8
9	Human COBRA 2	8
10	A/Sichuan/26211/2014	8
11	A/GYLFALCON/Washington/41088-6/2014	8
12	A/chivken/Egypt/CAL3-RLQP/2017	8
13	a/whooper swan/Mongolia/244/2005	8
14	Mock	8

## Data Availability

All data are available in the main text or the Appendix A.

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
