# Peer review of "Next-Generation Computationally Designed Influenza Hemagglutinin Vaccines Protect against H5Nx Virus Infections"

_pathogens, 2021, doi:10.3390/pathogens10111352_

Round 1

Reviewer 1 Report

In this study, Nunez et al. designed and tested computationally optimized broadly reactive H5 HA in mice for immunogenicity and cross-protection. Although the study is highly important for pandemic preparedness and the results are interesting, the presentation of the manuscript is very sloppy and full of errors that bewilder anyone who is reading it. 

Major issues:

  1. The authors must show the aa sequences of five COBRA H5 HA antigens used in the study.
  2. Explain what COBRA stands for at the beginning of the manuscript. 
  3. Fig 2 only shows 6 antisera and 5 viruses. Where is the 6th PR8-backbone virus? What does it mean "against a HAI panel of H5Nx viruses"?
  4. Similarly Fig 3 only shows 8 antisera and 5 viruses. Where are the data for the 6th, 7th and 8th PR8-backbone viruses? Did the authors intend to say assess the HAI response against a panel of H5Nx viruses?
  5. Fig 4 legend only explains two challenge viruses - VN/04 and Si/14. What about the other 3 viruses shown? What does it mean "viruses used were 6+2 recombinant, PR8-backbone isolates"? Were all five viruses 6:2 reassortants? Explain VN/04 and Si/14 abbrv in the figure legend. 
  6. Fig 5 is a disaster. I have never seen any paper that labels subfigures in B, A, D, C sequence. Also the Y axis scale of A & B is not "%" and needs to be corrected.
  7. Line 363-369: figure references are totally wrong. Where SI/14 was mentioned should be Figure 6B. Similarly where VN/04 was mentioned should be Figure 6A.
  8. The biggest issue is Fig 6B - absolutely no viral titers detected at dpi 3 after SI/14 challenge. Then how to explain IAN4 and IAN8 groups lost 10% BW after SI/14 challenge in Fig. 5B? Also how to explain the infiltration and inflammation in H&E staining of IAN4, IAN7 and IAN8 in Fig 7? Most of all, how to explain lung IHC staining in IAN4, IAN7 and IAN8 in Fig 9? If no lung viral titers were detected by plaque assay which is more sensitive than IHC staining, then what could cause inflammation and NP positive in those mouse lung tissues??? The authors must repeat plaque assay on lung homogenates after SI/14 challenge and revise corresponding results/discussion. 
  9. Fig 9 &10 legends: should be IHC staining, not H&E staining.

Author Response

Reviewer #1

Open Review

English language and style

(x) Extensive editing of English language and style required  
( ) Moderate English changes required  
( ) English language and style are fine/minor spell check required  
( ) I don't feel qualified to judge about the English language and style  

Yes

Can be improved

Must be improved

Not applicable

Does the introduction provide sufficient background and include all relevant references?

(x)

( )

( )

( )

Is the research design appropriate?

(x)

( )

( )

( )

Are the methods adequately described?

(x)

( )

( )

( )

Are the results clearly presented?

( )

( )

(x)

( )

Are the conclusions supported by the results?

( )

(x)

( )

( )

Comments and Suggestions for Authors

In this study, Nunez et al. designed and tested computationally optimized broadly reactive H5 HA in mice for immunogenicity and cross-protection. Although the study is highly important for pandemic preparedness and the results are interesting, the presentation of the manuscript is very sloppy and full of errors that bewilder anyone who is reading it. 

Major issues:

  1. The authors must show the aa sequences of five COBRA H5 HA antigens used in the study.
    • We’d like to thank the reviewer for the comment and have added the amino acid sequences to the supplemental figures
  2. Explain what COBRA stands for at the beginning of the manuscript. 
    • We have added the definition of COBRA to the manuscript.
  3. Fig 2 only shows 6 antisera and 5 viruses. Where is the 6th PR8-backbone virus? What does it mean "against a HAI panel of H5Nx viruses"?
    • The figure legend for Figure 2 has been updated to read “5 PR8-backbone viruses”. We describe a panel of viruses to be a list of viruses’ representatives of the numerous H5 clades and year isolates.
  4. Similarly Fig 3 only shows 8 antisera and 5 viruses. Where are the data for the 6th, 7th and 8th PR8-backbone viruses? Did the authors intend to say assess the HAI response against a panel of H5Nx viruses?
    • We have updated the figure legend to read “a panel of 5 PR8 backbone viruses”
  5. Fig 4 legend only explains two challenge viruses - VN/04 and Si/14. What about the other 3 viruses shown? What does it mean "viruses used were 6+2 recombinant, PR8-backbone isolates"? Were all five viruses 6:2 reassortants? Explain VN/04 and Si/14 abbrv in the figure legend. 
    • We have updated the materials and methods section to clearly explain that the viruses used are 6:2 PR8 reassortants. Abbreviations are also described in this section,
  6. Fig 5 is a disaster. I have never seen any paper that labels subfigures in B, A, D, C sequence. Also the Y axis scale of A & B is not "%" and needs to be corrected.
    • We would like to thank the reviewer for this positive feedback and have fixed the figure legend and apologize for the mixed labels.
  7. Line 363-369: figure references are totally wrong. Where SI/14 was mentioned should be Figure 6B. Similarly where VN/04 was mentioned should be Figure 6A.

    • The figure references have been corrected, thank you for bringing this to our attention.

  1. The biggest issue is Fig 6B - absolutely no viral titers detected at dpi 3 after SI/14 challenge. Then how to explain IAN4 and IAN8 groups lost 10% BW after SI/14 challenge in Fig. 5B? Also how to explain the infiltration and inflammation in H&E staining of IAN4, IAN7 and IAN8 in Fig 7? Most of all, how to explain lung IHC staining in IAN4, IAN7 and IAN8 in Fig 9? If no lung viral titers were detected by plaque assay which is more sensitive than IHC staining, then what could cause inflammation and NP positive in those mouse lung tissues??? The authors must repeat plaque assay on lung homogenates after SI/14 challenge and revise corresponding results/discussion. 
    • We would like to thank the reviewer for their feedback and discuss the results for Figure 6. For influenza-based assays, immunohistochemistry, or IHC, is a very sensitive assay for viral detection. As IHC uses monoclonal antibodies that bind specifically to viral nucleoprotein NP. Therefore, IHC can detect the presence of viral protein, whether there is live replicating virus or antigen being presented. In opposition to the common plaque assay which only detects live replicating virus. Therefore, the assay suggests that there was no replicating virus in the lungs following infection, however viral proteins were present. We believe that this represents a viral infection which was cleared 3 days post infection. This is also consistent with the weight loss of 10% seen in vaccinated mice.
    • Most importantly, our data is consistent with the weight loss pattern, where we see little to no weight loss in groups who are vaccinated, and extreme weight loss/death in the control groups. Therefore, we do not see a benefit to repeat the plaque assay as the data shows 1) mouse weight loss and death in control group, with little to no weight loss in vaccinated groups 2) high viral lung titers present in control mice lungs. Plaque assays were performed at the same time and same settings.

  1. Fig 9 &10 legends: should be IHC staining, not H&E staining.
    • This has been addressed, thank you for bringing it to our attention.

Submission Date

04 September 2021

Date of this review

15 Sep 2021 04:25:27

Reviewer 2 Report

The manuscript by Nunez et al highlights the use of bioinformatics to design novel vaccines against influenza viruses. The authors perform a thorough evaluation of a series of vaccine candidates based on the COBRA design and use an influenza mice model to examine virulence and efficacy. Overall, the authors provide a significant contribution in the field of influenza vaccines and the efforts toward the development of a universal influenza vaccine. The manuscript is well written, however, there are a series of inconsistencies and typos that  are indicated below and that the authors need to address:

Figure 1: the phylogenetic tree displays color-coded groups but no explanation is provided (i.e., red-vaccine HA sequence? vs purple vs black)

Although the authors are very familiar with the concept of using COBRA, I think the manuscript will benefit from defining COBRA: i.e., computationally optimized broadly reactive antigens. This will help younger investigators or other vaccinologists to understand the use of this technology.

The authors did not report the recombinant protein expression profiles of all the mammalian expression constructs evaluated in this manuscript. Only the methods are described but there is no evidence of the expression. I think it is important to display the protein expression profiles. Is it possible that certain post-translational modifications are detected (i.e., glycosylation) and are these having an impact on their biological activity?

Typos:

Line 37: remove “were”

Line 43: remove “in”

Line 63 : fix “deangth”

Line 146: fix “isoalted”

Line 193: fix “unites”

Line 323: fix “depeth”

Figure 4: it has letters B and C inside graphs for fig 4a and 4b.

Author Response

Reviewer #2

Open Review

English language and style

( ) Extensive editing of English language and style required  
( ) Moderate English changes required  
(x) English language and style are fine/minor spell check required  
( ) I don't feel qualified to judge about the English language and style  

Yes

Can be improved

Must be improved

Not applicable

Does the introduction provide sufficient background and include all relevant references?

(x)

( )

( )

( )

Is the research design appropriate?

( )

(x)

( )

( )

Are the methods adequately described?

(x)

( )

( )

( )

Are the results clearly presented?

(x)

( )

( )

( )

Are the conclusions supported by the results?

(x)

( )

( )

( )

Comments and Suggestions for Authors

The manuscript by Nunez et al highlights the use of bioinformatics to design novel vaccines against influenza viruses. The authors perform a thorough evaluation of a series of vaccine candidates based on the COBRA design and use an influenza mice model to examine virulence and efficacy. Overall, the authors provide a significant contribution in the field of influenza vaccines and the efforts toward the development of a universal influenza vaccine. The manuscript is well written, however, there are a series of inconsistencies and typos that are indicated below and that the authors need to address:

Figure 1: the phylogenetic tree displays color-coded groups but no explanation is provided (i.e., red-vaccine HA sequence? vs purple vs black)

  • We have updated the figure legend to represent the color coding being represented on the phylogenetic tree

Although the authors are very familiar with the concept of using COBRA, I think the manuscript will benefit from defining COBRA: i.e., computationally optimized broadly reactive antigens. This will help younger investigators or other vaccinologists to understand the use of this technology.

  • We would like to thank the reviewer for bringing this to our attention and have defined COBRA in the beginning of the manuscript.

The authors did not report the recombinant protein expression profiles of all the mammalian expression constructs evaluated in this manuscript. Only the methods are described but there is no evidence of the expression. I think it is important to display the protein expression profiles. Is it possible that certain post-translational modifications are detected (i.e., glycosylation) and are these having an impact on their biological activity?

  • All recombinant proteins are purified and quantified using Coomassie blue gel staining (not shown). The process of recombinant protein expression has been previously discussed in Ecker et al, 2020 (PMC7595037). Recombinant proteins produced in this manner have been found to have glycosylation patterns that are consistent with proteins produced by a human host. This has been validated by collaborators who have performed crystallography on COBRA H5 proteins (Bar-Peled, Yael et al, 2019; PM6736729). Influenza specific protein glycosylation are essential post-translational modifications in order to ensure proper folding of the protein and are therefore expressed in mammalian cell lines in order to closely replicate viral replication in a human cell line. COBRA-HA antigens are analyzed for specific N-linked glycosylation motifs are pre-predicted prior to expression.

Typos:

Line 37: remove “were”

Line 43: remove “in”

Line 63 : fix “deangth”

Line 146: fix “isoalted”

Line 193: fix “unites”

Line 323: fix “depeth”

Figure 4: it has letters B and C inside graphs for fig 4a and 4b.

  • Thank you for bringing these to our attention, the manuscript has been updated to reflect these typos.

Submission Date

04 September 2021

Date of this review

14 Sep 2021 16:04:39

Round 2

Reviewer 1 Report

  1. The authors must show the aa sequences of five COBRA H5 HA antigens used in the study.
    • We’d like to thank the reviewer for the comment and have added the amino acid sequences to the supplemental figures.

Where are the amino acid sequences of those 5 COBRA antigens? Supplementary Table 1 only lists the names of 5 H5 COBRA antigens and the ranges of sequences searched. No actual amino acid sequences are provided. The authors must provide the exact amino acid sequences of 5 H5 COBRA antigens that were used in the study. All materials used in the study should be fully disclosed to ensure the reproducibility of experiments. 

Fig. 6B, as the authors noted in the rebuttal letter that “IHC uses monoclonal antibodies that bind specifically to viral nucleoprotein NP”. This NP specific monoclonal antibody can also be used in cell-based ELISA for TCID50 assay. If plaque assay is not sensitive enough, then the authors should perform TCID50 to demonstrate no infectious viral particles present after vaccination, or the authors can conduct real-time PCR assays to show the overall viral loads.

Line 141: supplemental figure 1 is P-epitope analysis and is not about “the remaining 322 AA are used to create the CORBRA HA1 sequence”.

Figure S1 legend, where ae “51 amino acids Table A1”? Where is Table A1?

Supplemental or supplementary, make it consistent.

Figure S2E is the same as Figure 4A. Delete duplicate.

Figure 4 legend: “SI/14 (A), VN/04 (B), A/Eg/10 (C)”. What are about An/05 and Eg/07? D, E?

Line 331-333. “Mice vaccinated with IAN-7 HA had high HAI 331 antibody titers against clade 2.3.4.4 viruses (Si/14 ??? Fig.4A and Supplementary Fig 2) and moderate 332 HAI activity against An/05 and Eg/07 (Supplementary Figure 2)”. Supplemental figure S2 has no An/05 or Eg/07. Only Fig. 4 has An/05 (4D) or Eg/07 (4E).

Line 333-338: Grammatical errors in “Mice vaccinated with 333 IAN-8 HA had antibodies with an average HAI activity all viruses in the panel that were 334 greater than 1:80”. What did the authors mean “which was similar to the anti-335 bodies elicited by the VN/04 HA, as well as IAN4, IAN-7 and IAN-8 HA and the original 336 H-CO2 HA vaccines that did not induce antibodies with HAI activity against the ck/VN/08 337 virus”?

Line 356: “This weight loss was statistically the same as …” grammatical error.

Author Response

October 11, 2021

Dear Editor:  We hereby resubmit the manuscript # pathogens-1388995 with the title “Next-Generation Computationally Designed Influenza Hemagglutinin Vaccines Protect Against H5Nx Virus Infections”.

The authors appreciate the interest shown in our manuscript, and the constructive criticism and comments regarding our study. Please see our responses to editorial and reviewer comments below, along with a revised version of the manuscript with highlighted changes.

Comments

Comment #1:  Where are the amino acid sequences of those 5 COBRA antigens?  Supplementary Table 1 only lists the names of 5 H5 COBRA antigens and the ranges of sequences searched. No actual amino acid sequences are provided. The authors must provide the exact amino acid sequences of 5 H5 COBRA antigens that were used in the study. All materials used in the study should be fully disclosed to ensure the reproducibility of experiments.

Response:

  • We would like to thank the reviewer for their comment and have added the supplemental figure to the manuscript containing all 8 COBRA sequences.

Comment #2:  Fig. 6B, as the authors noted in the rebuttal letter that “IHC uses monoclonal antibodies that bind specifically to viral nucleoprotein NP”. This NP specific monoclonal antibody can also be used in cell-based ELISA for TCID50 assay. If plaque assay is not sensitive enough, then the authors should perform TCID50 to demonstrate no infectious viral particles present after vaccination, or the authors can conduct real-time PCR assays to show the overall viral loads.

Response:

  • We would like to thank the reviewer for assessing this figure carefully. TCID50 tests inherently involve viral infection and cellular cytopathic effects (CPE) which is also seen in viral plaque assays. Since the control mice did in fact, display viral lung plaques and suffered severe weight loss and death compared to the vaccinated mice, we can conclude that the vaccines provided protection against infection enough to inhibit viral loads in the lungs.

Comment #3:   Line 141: supplemental figure 1 is P-epitope analysis and is not about “the remaining 322 AA are used to create the CORBRA HA1 sequence”.

Response:

  • The manuscript has been edited to reflect these changes

Comment #3:   Figure S1 legend, where ae “51 amino acids Table A1”? Where is Table A1? Supplemental or supplementary, make it consistent.  Figure S2E is the same as Figure 4A. Delete duplicate.  Figure 4 legend: “SI/14 (A), VN/04 (B), A/Eg/10 (C)”. What are about An/05 and Eg/07? D, E?

Response:

  • The manuscript has been edited to reflect these comments.

Comment #4:   Line 331-333. “Mice vaccinated with IAN-7 HA had high HAI 331 antibody titers against clade 2.3.4.4 viruses (Si/14 ??? Fig.4A and Supplementary Fig 2) and moderate 332 HAI activity against An/05 and Eg/07 (Supplementary Figure 2)”. Supplemental figure S2 has no An/05 or Eg/07. Only Fig. 4 has An/05 (4D) or Eg/07 (4E).

Response:

  • We would like to thank the reviewer for pointing out this inconsistency and we have changed the manuscript to reflect the correct figure.

Comment #5:   Line 333-338: Grammatical errors in “Mice vaccinated with IAN-8 HA had antibodies with an average HAI activity all viruses in the panel that were greater than 1:80”. What did the authors mean “which was similar to the antibodies elicited by the VN/04 HA, as well as IAN4, IAN-7 and IAN-8 HA and the original 336 H-CO2 HA vaccines that did not induce antibodies with HAI activity against the ck/VN/08 337 virus”?

Line 356: “This weight loss was statistically the same 1:80s …” grammatical error.

Response:

  • Thank you for bringing this to our attention, the manuscript has been updated to reflect these changes.

Best regards,

Ted M. Ross, Ph.D.

GRA Eminent Scholar in Infectious Diseases

Director - Center for Vaccines and Immunology

Professor - Department of Infectious Diseases

University of Georgia

Round 3

Reviewer 1 Report

No further comments.